# The Steric Effect in Green Benzylation of Arenes with Benzyl Alcohol Catalyzed by Hierarchical H-beta Zeolite

**Xinyu Liu [†], Meihuan Lu [†], Xuan Wang, Juyou Lu and Jianxin Yang ***

Hainan Provincial Fine Chemical Engineering Research Center, Laboratory of Green Catalysis and Reaction Engineering of Haikou, Key Laboratory of Tropical Biological Resources of Ministry of Education, College of Science, Hainan University, Haikou 570228, China; liuxyhainu@163.com (X.L.); lumhuan@163.com (M.L.); wangxuan2788@163.com (X.W.); lujy@hainanu.edu.cn (J.L.)
* Correspondence: yangjxmail@hainanu.edu.cn; Tel.: +86-898-66279161
† These authors contributed equally to this work.

**Abstract:** For decades the steric effect was still ambiguously understood in catalytic benzylation reactions of arenes with benzyl alcohol, which limited the green synthesis of phenylmethane derivates in industrial scale. This research applies a series of silica–alumina beta zeolites to systematically evaluate factors like catalyst porosity, reactants molecule size, and reaction temperature on catalytic benzylation. First, a suitable hierarchical beta zeolite catalyst was screened out by X-ray powder diffraction, $N_2$ adsorption−desorption, and probe benzylation with *p*-xylene. In the following substrates expanding study, for a typical benzylation of benzene, it showed extraordinary performance among literature reported ones that the conversion was 98% while selectivity was 90% at 353 K only after 10 min. The steric effect of aromatics with different molecular sizes on benzylation was observed. The reaction activities of four different aromatics followed the order: benzene > toluene > *p*-xylene > mesitylene. Combined with macroscopic kinetic analysis, this comprehensive study points out for the first time that the nature of this steric effect was dominated by the relative adsorption efficiency of different guest aromatic molecules on the host zeolite surface.

**Keywords:** benzylation; benzyl alcohol; hierarchical zeolite; steric effect; green catalysis

---

## 1. Introduction

Friedel–Crafts alkylation has been an important method for preparing substituted aromatics from the past to the present days [1–3]. Among these classical, but practical alkylation reactions, the homogeneous benzylation of aromatic hydrocarbons by benzyl alcohol (BzOH) or benzyl chloride (BzCl) is one of the significant reaction processes for the preparation of diphenylmethane (DPM) and its substituted derivatives (Scheme 1). As important fine chemical intermediates, DPM and its derivatives are widely used in industry such as making artificial flavors, fragrance, medicine, dyes, and heat conduction media [4]. Compared to BzCl, which produces HCl as a by-product and, often causes corrosion and catalyst leaching problems, BzOH is an environmentally friendly benzyl reagent, whose main by-product besides benzylation is only water. However, in the research for better reactivity and no etherification to form dibenzyl ether (DBE), to decrease selectivity [5], extensive works have already been published using BzCl [6–11]. It is still necessary to gain more insight on understanding catalytic reaction mechanisms with the more challenging BzOH as benzyl reagent because as one of the enduring research themes in chemistry [12–18], developing green catalytic process is beneficial to build an ecofriendly and sustainable economy.

**Scheme 1.** Procedures of the benzylation of aromatics with benzyl alcohol. (**a**) Ar = phenyl; (**b**) Ar = tolyl; (**c**) Ar = xylyl; (**d**) Ar = mesityl.

Generally, benzylations are carried out under homogeneous catalysts such as $H_2SO_4$, $ZnCl_2$, $AlCl_3$, $FeCl_3$, $ZnCl_2$, and $BF_3$, which brings some negative aspects, such as uncontrollable multi benzylation products (polymers in Scheme 1), toxicity, corrosion, and difficulties in the recovery of catalyst [9]. Therefore, developing an excellent heterogeneous acid catalyst has been a challenging issue for a long time. Some valuable publications showed that supported metal catalysts and heteropoly acid catalysts has good properties in the catalytic benzylation of arene with benzyl chloride or benzyl alcohol, but many problems remain due to the loss of active components and relatively high cost [9,19,20]. Consequently, a category of silica–alumina zeolite with unique pore system, large surface area, better stability and acidity has attracted considerable interest [21–25].

In recent years, most research has focused on preparing zeolites like ZSM-5 [22,23], mordenite [24,25] and beta [5,26–28] with new methods and exploring the potential application in the Friedel–Crafts reaction. Among them, beta zeolite has received considerable interests. Kim et al. reported that nano beta zeolite deactivated more slowly in liquid phase Friedel–Crafts benzyl process [26]. Candu et al. compared H-beta with mordenite zeolite in the benzylation of benzene with benzyl alcohol with three experiment methodologies and they found H-Beta zeolite was a better candidate [29]. The hierarchical beta zeolite consists of straight 12-membered ring channels, and the introduction of mesopore system makes the mass transfer even better for large molecules [30]. Additionally, many valuable research works [30–34] have provided green methods to prepare hierarchical beta zeolite. The Thomas Bein group classified the preparation methods of hierarchical zeolite into six categories [35].

Herein, our work is a comprehensive study concentrated on the benzylation procedures of different arenes with benzyl alcohol by employing hierarchical H-beta zeolite. The preparation of catalysts was done according to two representative ways, "bottom-up" and "top-down" in hierarchical zeolite design [32]. The suitable catalyst candidate was screened out by essential characterizations and probe reaction. The catalytic properties of the H-beta catalyst were investigated by the benzylation of different sizes of aromatics such as benzene, toluene, *p*-xylene and mesitylene with benzyl alcohol. Based on kinetic parameter analysis, the reaction mechanism was also investigated. Interestingly, the steric effect of molecular size of arenes on performance was also observed. For the first time, the ambiguous steric effect in benzylation was linked to adsorption ratio of reactants with kinetic evidence.

## 2. Results and Discussion

### 2.1. Essential Characterization of Beta Zeolite Catalysts

The powder X-ray diffraction pattern showed the characteristic peaks of the obtained beta zeolite samples (Figure 1). Two dominant peaks appeared in the two-theta region = 7.6°(101) and 22.4° (302), indicating that all the samples have microporous BEA structure as described in the literature [31] and

JCPDS cards 48-0038. Hβ-1 was assigned a crystallinity of 100%. The relative crystallinity of Hβ-2 decreased to 57% because the skeleton was damaged after base treatment, and the crystallinity of Hβ-3 was recovered to 75% after acid treatment, which is in agreement with the description in Ref [28]. However, the relative crystallinity of Hβ-4 was only up to 48%, perhaps because our experiment condition is different from that in ref [30]. The nitrogen adsorption isotherms exhibited the texture properties of all the catalysts (Figure 2, Table 1). Hβ-2, Hβ-3 and Hβ-4 gave isotherm of model I at low relative pressure in the range of $P/P_0 < 0.10$ and at high relative pressure gave isotherm of model IV with a hysteresis loop in the range of $0.60 < P/P_0 < 0.90$, indicating the coexistence of micropores and mesopores, in comparison to the microporous sample Hβ-1 which only gave the isotherm of model I.

It is worth to note that t-plot method was pointed out to underestimate the microporous volume for micro/meso materials and BJH method was thought not as precise as NLDFT method in measuring pore diameter [36,37]. As a typical probe molecule in physical adsorption, diatomic molecule $N_2$ was questioned for long shape and large quadrupole moment, making it not a perfect choice compare to Ar. Although these widely used adsorption method might be imperfect, the introduced system errors were presumed not affect the confirmation of existing hierarchical pore structure for asprepared samples.

In a word, these essential characterizations confirmed the hierarchical beta zeolite prepared here, paving the way for this applied catalysis study which focuses on studying catalytic reaction.

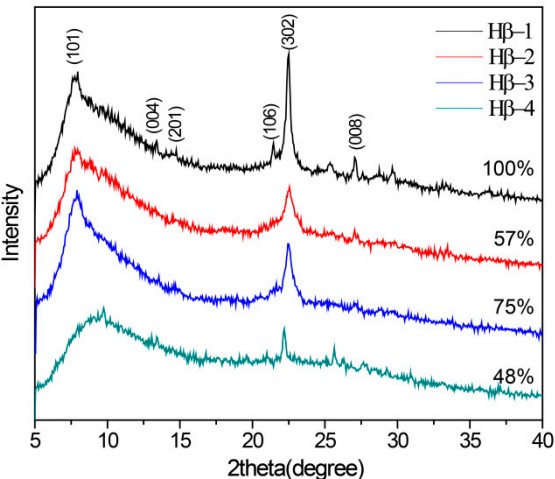

**Figure 1.** X-ray powder diffraction (XRD) patterns Hβ-1 (black), Hβ-2 (red), Hβ-3 (blue) and Hβ-4 (green).

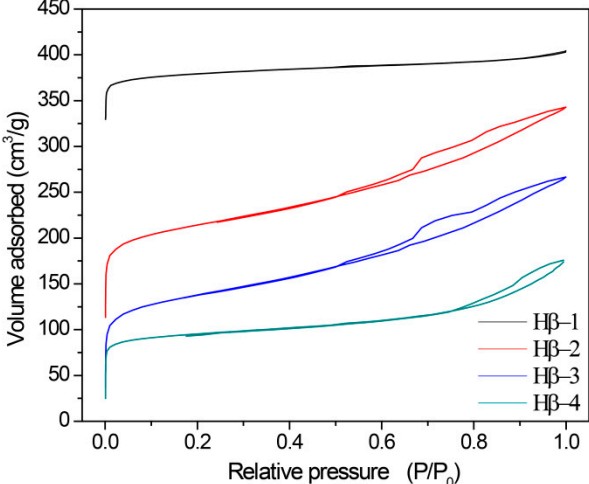

**Figure 2.** Nitrogen adsorption and desorption isotherms of the catalysts.

**Table 1.** Textural properties and chemical composition of the catalysts.

| Sample | BET Surface Area (m²/g) [a] | External Surface Area (m²/g) [b] | Micropore Volume (cm³/g) [b] | Mesopore Volume (cm³/g) [c] | Mesopore Diameter (nm) [d] | Si/Al [e] |
|---|---|---|---|---|---|---|
| Hβ-1 | 345.7 | 63.6 | 0.14 | 0.06 | - | 25.0 |
| Hβ-2 | 366.5 | 212.0 | 0.07 | 0.30 | 6.1 | 21.4 |
| Hβ-3 | 375.6 | 215.9 | 0.08 | 0.30 | 6.3 | 23.7 |
| Hβ-4 | 241.5 | 76.9 | 0.08 | 0.16 | 7.3 | 28.4 |

[a] BET multiplot method. [b] t-Plot method. [c] BJH method (adsorption branch), the measured intercrystalline mesopore diameter of microporous Hβ-1 was omitted. [d] NLDFT method. [e] XRF.

## 2.2. Benzylation of Arenes with Benzyl Alcohol

The screening of catalysts mentioned above was conducted by a probe benzylation of *p*-xylene, because *p*-xylene showed a relatively moderate molecule size and the lowest biological toxicity of all the aromatics used in this work. As depicted in Figure 3, Hβ-3 exhibited the best catalytic properties with the reaction conditions: catalyst (50 mg), *p*-xylene (4 mL) and benzyl alcohol (0.25 mL), 373K, reaction time 4 h.

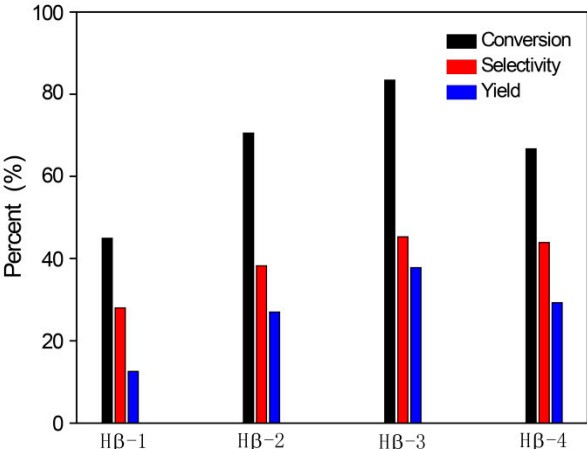

**Figure 3.** The performance of the catalysts in a probe benzylation of *p*-xylene. (The conversion was based on benzyl alcohol, the selectivity and yield referred to the main benzylation product 2-benzyl-1,4-dimethylbenzene).

To give possible explanations for such catalytic performance of different samples, structural factors were briefly discussed. Acidity usually plays an important role in solid acid catalysis. Here NH$_3$-TPD results show the acid amount of different acid strength in Figure 4a, all the beta zeolites are similar in acid strength distribution with weak acid sites and strong acid sites around 453 K and 573K, respectively. Microporous Hβ-1 shows largest total acid amount than other hierarchical beta zeolites, which agrees well with the relative crystallinity from X-ray powder diffraction (XRD) results. To obtain more detailed information about Brönsted acid sites (BAS) and Lewis acid sites (LAS), the amount of weak acid sites from NH$_3$-TPD results was used to represent amount of LAS, while the amount of strong acid sites from NH$_3$-TPD results was used to represent amount of BAS [36,37]. A correlation of catalytic activity with the amount of BAS and LAS was built in Figure 4b, from which we can see in general that in the probe benzylation of *p*-xylene in 373K, three hierarchical beta zeolites Hβ-4, Hβ-2 and Hβ-3 show rising activity as the BAS and LAS amount increase, in consistent with previous studies in catalytic benzylation [5,7,28]. However, microporous Hβ-1 is superior in BAS and LAS amount but has the lowest activity, and we infer the less external surface area of Hβ-1 (Table 1) was to blame. It is already known that porosity and acidity both affect the catalytic performance in benzylation reactions with zeolite catalysts [7,11,28]. Wang et al. [28] studied the difference of parent beta zeolite

and mesoporous zeolite made from post treatment method in catalyzing benzylation of benzene and mysitylene. By using pivalonitrile-FTIR and NMR techniques, they specially highlighted the importance of accessible acid sites. According to their view, small base probe molecules like $NH_3$ and pyridine fail to access the external surface, which is available for large reactant molecule. Hao et al. [7] also found the accessibility of BAS is critical to the activity and both BAS and LAS have positive correlation with catalytic activity in benzylation of naphthalene. To sum up, combine acidity and porosity, containing more abundant acid sites amount than Hβ-2 and Hβ-4, and larger external surface area than microporous Hβ-1, no wonder Hβ-3 shows the best performance.

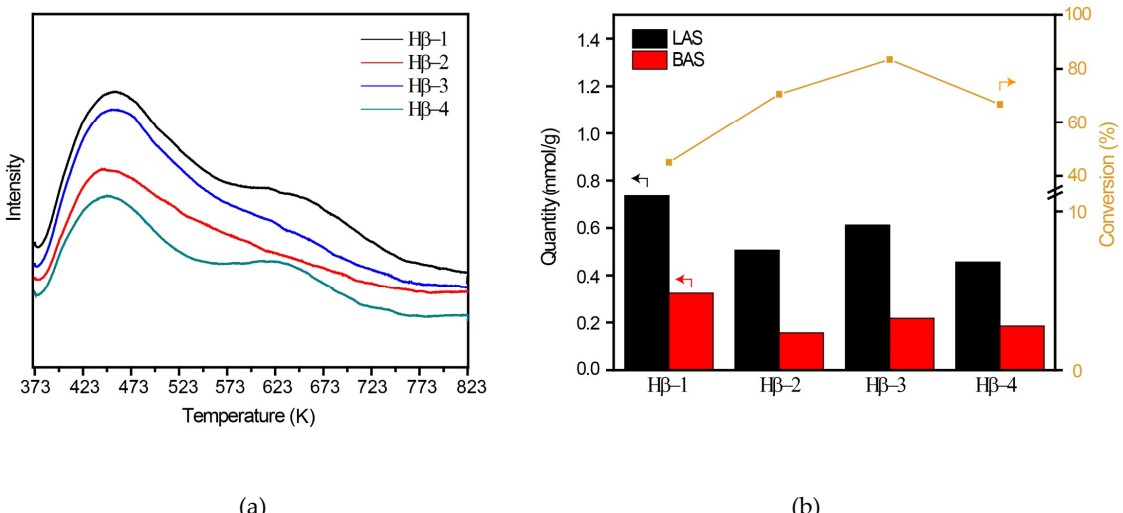

(a)                                   (b)

**Figure 4.** (**a**) $NH_3$-TPD profiles of samples from Hβ-1 to Hβ-4. (**b**) The correlations between different acidity site amounts, possible Brönsted acid sites (BAS) and Lewis acid sites (LAS) with catalytic activity in benzylation of *p*-xylene. The conversion was based on benzyl alcohol.

Therefore, Hβ-3 was selected as an optimal catalyst for the following benzylation study of aromatic hydrocarbon with benzyl alcohol. It is worthy to note that in order to link the reaction results with the intrinsic kinetics of main reaction, a larger feeding ratio of aromatics (7 mL) to benzyl alcohol (0.1 mL) was employed in following sections of this article, which is reported as an effective way to improve the selectivity of mono benzylation products [29].

### 2.2.1. Benzylation of Benzene with Benzyl Alcohol

As the simplest aromatic hydrocarbon, the symmetrical six-member ring without substituents makes benzene an unideal candidate for electrophilic substitution. Nevertheless, in this work benzylation of benzene with benzyl alcohol took place easily catalyzed by Hβ-3, at 353 K. The product detected by Gas Chromatography-Mass Spectrometer (GC-MS) was DPM, as well as DBE which are shown in Scheme 1a. The conversion of benzyl alcohol, selectivity and yield of target product DPM are shown in Figure 5. The catalyst showed excellent activities that the conversion of benzyl alcohol rose rapidly to 98% only at a time of 10 min, the highest efficiency among the ever reported ones. Meanwhile, the yield of DPM rose to almost 90% very quickly when the intermediate product DBE converted to DPM completely.

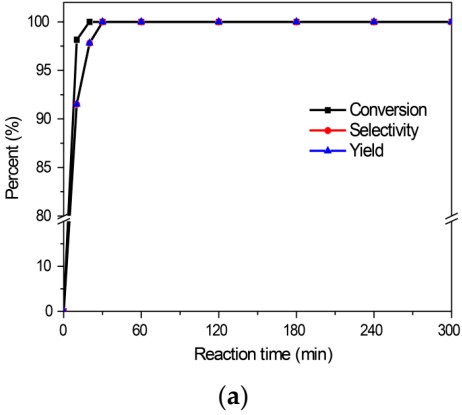 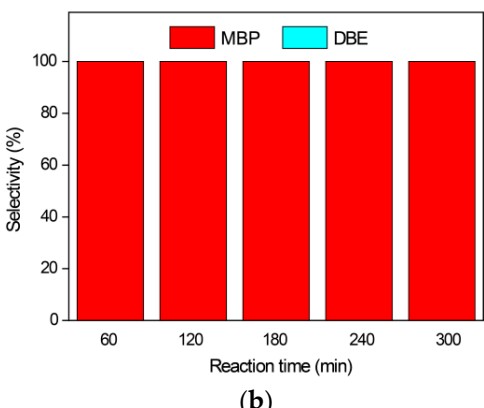

(**a**)　　　　　　　　　　　　　　(**b**)

**Figure 5.** Benzylation reaction of benzene and benzyl alcohol at 353 K. (**a**) Conversion (based on benzyl alcohol), selectivity and yield (referred to diphenylmethane). (**b**) Product distribution of monobenzylation products (MBP, diphenylmethane in this case) and dibenzyl ether (DBE).

### 2.2.2. Benzylation of Toluene with Benzyl Alcohol

When employing toluene and benzyl alcohol as raw materials at 373 K, the mainly products were benzyl-4-methyl benzene (4-BMB), benzyl-2-methylbenzene (2-BMB) and an intermediate DBE as shown in Scheme 1b. The ratio of the three main mono benzylation products (4-BMB, 2-BMB and 3-BMB) was 9.0:2.6:1. The mono benzylation products were regarded as the target product for selectivity calculations. The conversion of benzyl alcohol, selectivity and yield of target product are shown in Figure 6. Again, the catalyst showed excellent activities in benzylation of toluene with benzyl alcohol. The conversion of benzyl alcohol increased rapidly and reached almost 98% at 30 min under Hβ-3. While the yield of products increased quickly and reached 92% after reacting for 30 min monitored by GC-MS.

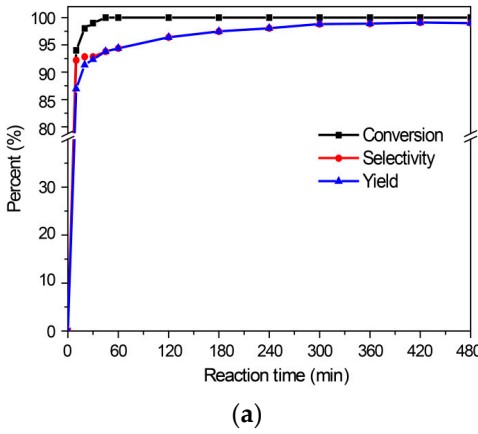 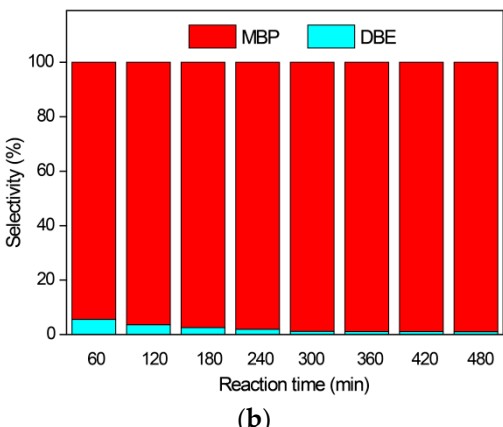

(**a**)　　　　　　　　　　　　　　(**b**)

**Figure 6.** Benzylation reaction of toluene and benzyl alcohol at 373 K. (**a**) Conversion (based on benzyl alcohol), selectivity and yield (referred to diphenylmethane). (**b**) Product distribution of mono benzylation products (MBP, benzyl-2-methylbenzene in this case) and DBE.

### 2.2.3. Benzylation of P-Xylene and Mesitylene with Benzyl Alcohol

Benzylations of *p*-xylene and mesitylene were similar to the former procedures. When using *p*-xylene as raw materials, the products were mainly 2-benzyl-1,4-dimethylbenzene (BDB) and intermediate DBE as shown in Scheme 1c. The conversion of benzyl alcohol, selectivity and yield of target product were shown in Figure 7. It is obvious that the conversion increased to 92% while the yield upgraded to 80% at 30 min, and the selectivity and yield of BDB reached almost 100% at 5 h until the intermediate DBE was completely converted to the desired product BDB identified by GC-MS.

By contrast, employing mesitylene as raw material, the main product was 2-benzyl-1,3,5-trimethylbenzene (BTB) and intermediate DBE (Scheme 1d). As shown in Figure 8, the conversion was only 68% while the yield was still less than 40% at 30 min. As the reaction continued, the selectivity was over 90% and yield of BTB reached almost 95% at 5 h until the intermediate DBE was completely converted to desired product BTB identified by GC-MS.

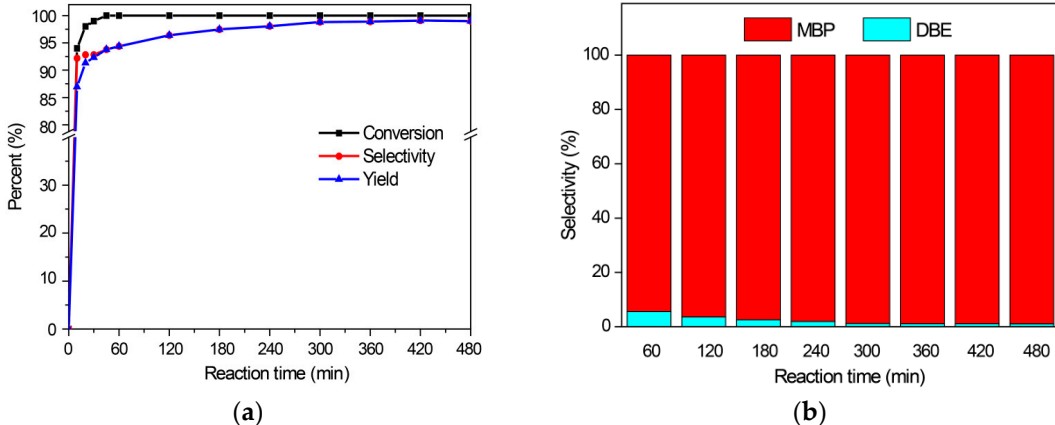

**Figure 7.** Benzylation of *p*-xylene and benzyl alcohol at 373 K. (**a**) Conversion (based on benzyl alcohol), selectivity and yield (referred to diphenylmethane). (**b**) Product distribution of mono benzylation products (MBP, 2-benzyl-1,4-dimethylbenzene in this case) and DBE.

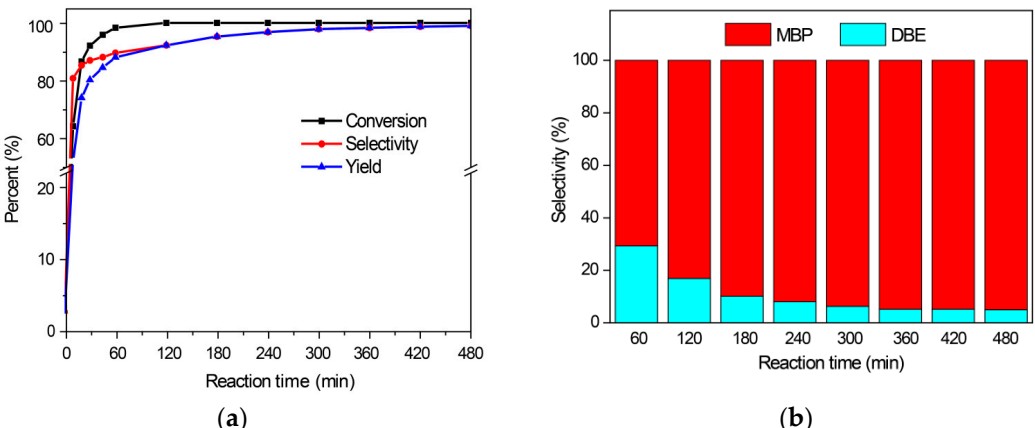

**Figure 8.** Benzylation of mesitylene and benzyl alcohol at 373 K. (**a**) Conversion (based on benzyl alcohol), selectivity and yield (referred to diphenylmethane) (**b**) Product distribution of mono benzylation products (MBP, 2-benzyl-1,3,5-trimethylbenzene in this case) and DBE.

### 2.2.4. The Reaction Activities of Different Arenes Substrates

The results of the benzylations catalyzed by Hβ-3 were summarized in Table 2. With the increasing of molecular size of arenes, conversion and selectivity of each reaction decreased, and the apparent rate constant $k_a$ descended obviously. According to the obtained apparent rate constant of each reaction, it should be concluded that the activities of different aromatics follow the order: mesitylene < *p*-xylene < toluene < benzene, which is consistent with the reported alkylation reaction over mesoporous Al-SBA-15 [38]. The above result was not in accordance with the traditional Friedel-Crafts alkylation, which believes that the existing of electron-donating groups make the reaction easier, so the activity order should be mesitylene > *p*-xylene > toluene > benzene [1]. Obviously, our results were different, and we assume that the diffusion and adsorption steps might be responsible. Considering the porous structures of hierarchical zeolites, when the molecular diameter of the aromatic molecule becomes larger, it became more difficult to access the internal surface of the catalyst. Therefore, the activity

order might be determined by the molecular steric hindrance as well as electron density. The former will be discussed in this article.

**Table 2.** Benzylation of aromatics and benzyl alcohol catalyzed by Hβ-3.

| Entry | Substrates | Temperature/K | a × b × c /nm × nm × nm [39] | Conversion/% | Selectivity/% | $k_a$/min$^{-1}$ |
|---|---|---|---|---|---|---|
| 1 | | 353 | 0.341 × 0.666 × 0.729 | 100 | 100 | 0.3993 |
| 2 | | 373 | 0.380 × 0.666 × 0.818 | 100 | 94 | 0.1191 |
| 3 | | 373 | 0.390 × 0.666 × 0.919 | 98 | 89 | 0.0566 |
| 4 | | 373 | 0.425 × 0.829 × 0.888 | 92 | 74 | 0.0477 |

Reaction conditions: Hβ-3 (50 mg), aromatics (7 mL) and benzyl alcohol (0.1 mL), 60 min. "a × b × c" means three perpendicular directions at minimum energy conformation when describing molecular size. The conversion was based on benzyl alcohol, while the selectivity and yield referred to corresponding mono benzylation products. ka is apparent rate constant, and the calculation was described in 2.3 in details.

## 2.3. Effects of Reaction Temperature

Temperature had a strong impact on both conversion and the selectivity in the benzylations (Figures 9 and 10). With the increase of temperature, the conversion and selectivity increased fast in a typical reaction of benzyl alcohol with excess of mesitylene. While the reaction temperature was at 333 K, benzylation did not take place. There was only a small quantity of etherification from benzyl alcohol, and no target product BTB after 6 h. When the temperature increased to at 353 K, benzyl alcohol was consumed slowly and the selectivity of target product BTB reached to 68% after 6 h; when the temperature increased to 373 K, benzyl alcohol was consumed almost completely and the selectivity of target product BTB reached to 86% after 2 h; and at 393 K, benzyl alcohol was consumed rapidly and the selectivity of target product BTB reached 95 % after 30 min. As mesitylene was in excess, the reaction fitted to a pseudo-first order reaction equation: $\ln (1/1-x) = k_a (t-t_o)$, in which $k_a$ is the apparent rate constant, $x$ is the conversion of benzyl alcohol, $t$ is the corresponding sampling time and $t_o$ is the induction period which contains metastable states like the system reach a stable temperature. A linear graph $\log (1/1-x)$ versus $t-t_o$ was acquired according to the pseudo-first order reaction equation (Figure 11). As it depicted, the estimated apparent activation energy was 169.10 kJ/mol (Figure 12).

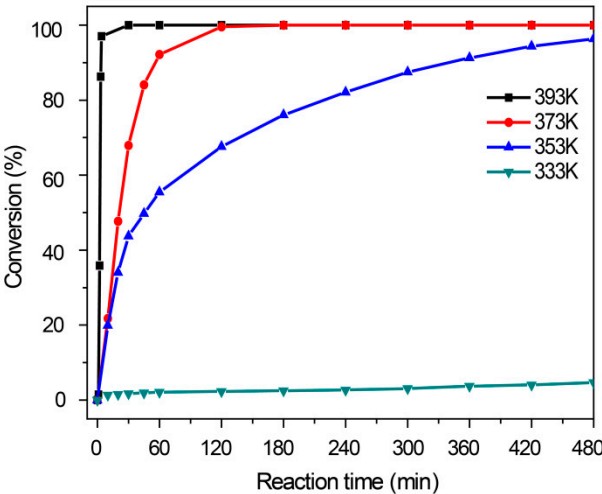

**Figure 9.** Conversion of benzyl alcohol in benzlyation of mesitylene.

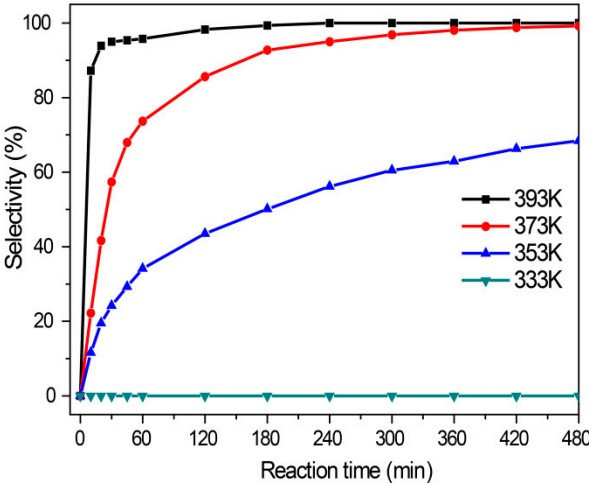

**Figure 10.** Selectivity of target product in benzlyation of mesitylene.

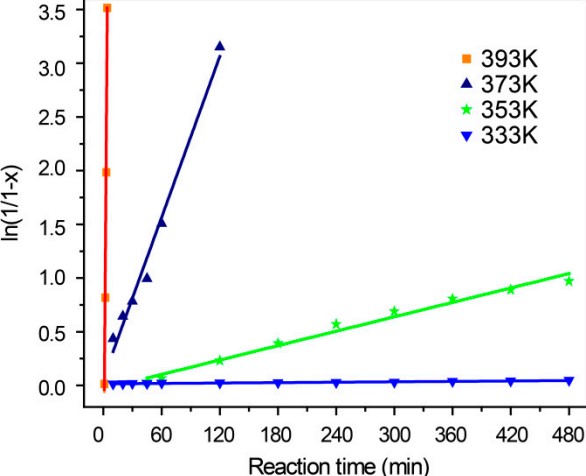

**Figure 11.** ln (1/1−x) versus reaction time in different temperature.

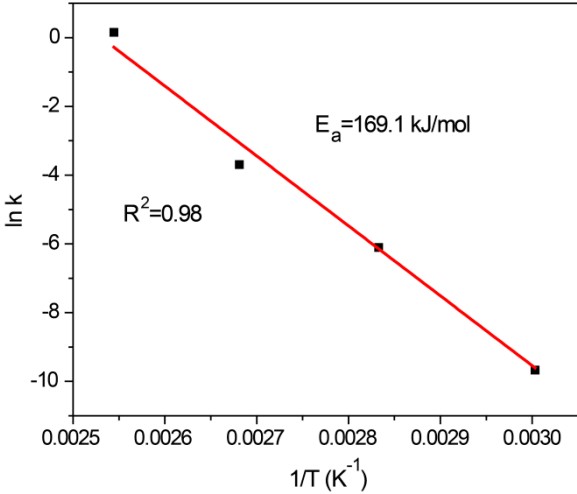

**Figure 12.** The linear Arrhenius graph of benzylation of mesitylene on Hβ-3.

*2.4. Recycling of the Catalyst*

Good reusability is an advantage of zeolite as a green heterogeneous catalyst in benzylation comparing to state of art $AlCl_3$ catalyst. After reaction, the zeolite was filtered and washed with ethanol followed by dried and calcined at 823 K for 3 h prior to reuse. Table 3 shows the reusability of initial and recycled Hβ-3. The initial catalysts gave 98% of benzyl alcohol conversion and 85% of 2-benzyl-1,4-dimethylbenzene selectivity. After five cycles, the catalytic property was still efficient that the conversion declined to 88% while the selectivity declined to 82%. Probably the decrease in activity was due to the loss of catalyst during filtering procedures (only more than 90% catalysts could be collected in Cycle 4 and 5).

**Table 3.** Reusability of Hβ-3 in benzylation of *p*-xylene.

| Items | Initial | Cycle 1 | Cycle 2 | Cycle 3 | Cycle 4 | Cycle 5 |
|---|---|---|---|---|---|---|
| Conversion (%) | 98 | 97 | 96 | 94 | 90 | 88 |
| Selectivity (%) | 85 | 80 | 82 | 81 | 82 | 82 |

Reaction conditions: Hβ-3 (50 mg), *p*-xylene (7 mL), benzyl alcohol (0.1 mL), 373 K, 1 h. (The conversion was based on benzyl alcohol, while the selectivity and yield referred to 2-benzyl-1,4-dimethylbenzene).

*2.5. Mechanism Discussion*

2.5.1. The Whole Heterogeneous Catalytic Process

For a typical heterogeneous process, the whole catalytic procedures can be divided into five steps (left flow chart in Figure 13). Benzyl alcohol and aromatic hydrocarbon molecules diffuse into the external and internal surface of zeolites, and then adsorption happens to make reactants bind on active sites for surface reaction. The products undergo desorption and diffuse out from internal and external surface consecutively.

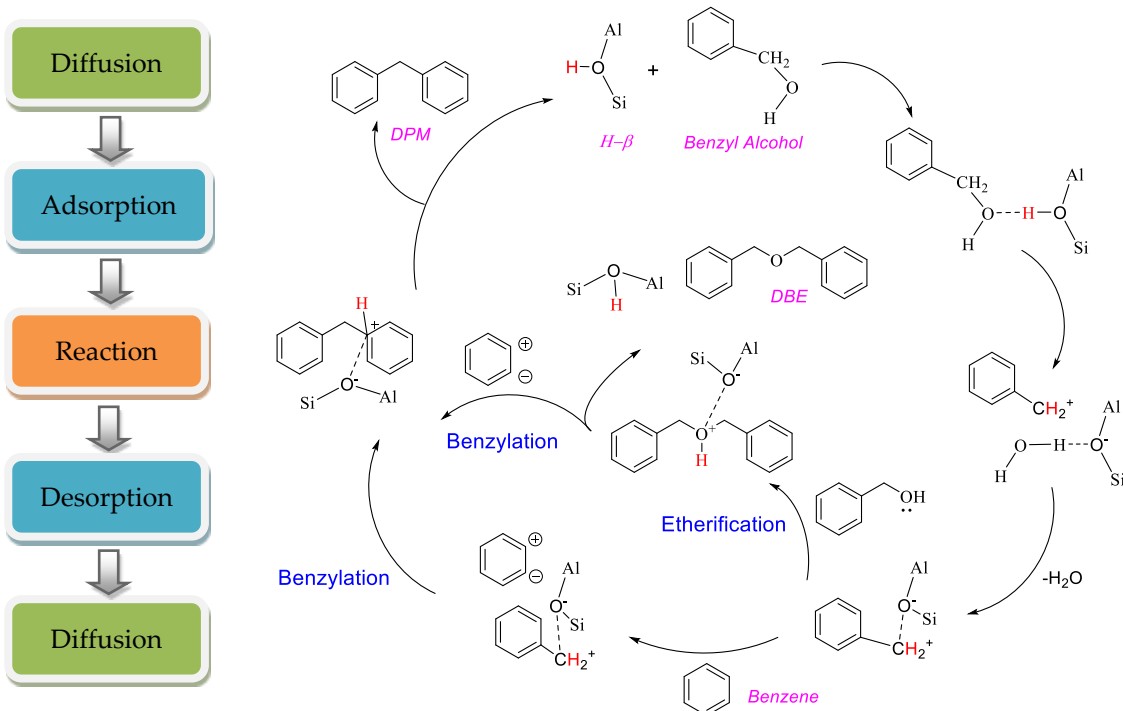

**Figure 13.** Inferred mechanism of main heterogeneous catalysis process (**left**). The reaction pathway of benzylation and etherification (**right**).

Among the above five steps, we speculate the surface reaction pathway (right cyclic graph in Figure 13) based on the viewpoint proposed by Narender et al. [2]. Take benzylation of benzene as an example, benzyl cation ($C_6H_5CH_2^+$) was initially appeared when hydroxyl oxygen forms hydrogen bond with Brønsted acid sites. Two competitive reactions coexist in the whole benzylation process. One is the benzyl cation attacking the activated aromatic ring to produce DPM directly; the other is the benzyl cation attacking electron pair of oxygen from benzyl alcohol to form DBE, which consecutively reacts with arenes to obtain DPM. The regeneration of "H" on zeolite comes from unstable protonated benzene ring to make the final product free of charge. The next cycle will start again.

To explore the kinetic nature of steric effect about different aromatics (results of $K_a$ in Table 2), representative steps in heterogeneous catalysis were discussed respectively in following sections.

2.5.2. Diffusion

Herein, the diffusion results were analyzed combining data from literature. Intracrystalline (configurational) diffusion is known as the dominant diffusion procedure in zeolites; this intracrystalline diffusion shows higher energy barrier than molecular and Knudsen diffusion [40–42]. The Corma group [43] once studied diffusion coefficients combined with experimental and theoretical methods. By infrared spectrometry, they measured the diffusion energy followed the order: benzene < toluene < *o*-xylene < ethylbenzene. It is harder for the molecule with a large kinetic diameter to jump between adsorption sites on zeolite surface according to transitional state theory. Similar results were obtained in other type of zeolites by ZLC method [44] or molecule dynamic simulations [45,46].

First of all, the effect of external mass transfer was estimated by Mears' criteria [39]:

$$C_M = \frac{-r'_A \rho_b R_n}{k_c c_{Ab}}, \tag{1}$$

where $r'_A$, observed reaction rate for typical benzylation of benzene, which is no more than the initial rate, as 0.0124 mol/ (L·min). $\rho_b$ is the bulk density of catalysts, cited as 1560 kg/m³ [47]. $R_n$ is particle

sizes, $k_C$ means liquid−solid external mass transfer coefficient, estimated as $1.36 \times 10^3$ m$^3$/(g·s) [47]. $C_{Ab}$ is the concentration of BzOH, which is 0.14 mol/L in this work. The calculated value meets: $C_M < 0.15$ and the impact of external diffusion is not dominated. This is consistent with the test results reported by Beltrame et. al [48,49], who thought high speed stirring as 1000 r/min during benzylation eliminated the impact of external mass transfer.

Then, the effect of internal mass transfer was estimated by Weisz–Prater' criteria [39]:

$$C_{wp} = \frac{-r'_A \rho_{cat} R_n}{D_{eff} c_{As}}, \tag{2}$$

$\rho_{cat}$ is the density of catalysts, given as 1000 kg/m$^3$ [47]. $D_{eff}$ is the effective diffusion coefficient whose order of magnitude is approximately evaluated to $10^{-9}$ m$^2$/s [50]. $C_{As}$ is the surface concentration of BzOH, whose order of magnitude was once measured as 1 mol/L [51]. The calculated results meet: $C_{wp} < 1$.

$$\Phi = L \sqrt{\frac{k}{D}}. \tag{3}$$

Thiele modulus is a dimensionless parameter, commonly used as measuring of the ratio of surface reaction rate to pore diffusion rate, which is shown in Equation (3). Taken L as diffusion distance (sometimes particle radius was used [52]), and for beta zeolite, the particles size was 100 nm (from Nankai University Catalyst Plant). The measured k = 0.3993 min$^{-1}$ and coefficient D = $3.2 \times 10^{-9}$ cm$^2$/s [43]. So, $\Phi$ is estimated as 0.86 < 1. Therefore, combining the $C_{wp}$ results with Thiele Modulus, the impact of internal diffusion is not important. Even when not decisive in the whole catalytic process, we could believe that the four guest molecules share different kinetics in the diffusion process on host Hβ-3, which is a reflection of steric effect.

### 2.5.3. Adsorption and Surface Reaction

Langmuir–Hinshelwood mechanism can be used to study the kinetics in liquid phase benzylations of arenes over solid acid catalysts [53], which describe the process as two reactant molecules absorb on catalysts surface respectively, followed by surface reaction, which can be summarized as Equation (4) in this case,

$$r = \frac{k_0 C_{cat} C_{aro} C_{bzh} C_P}{(1 + K_{aro} C_{aro} + K_{bzh} C_{bzh} + K_P C_P)^2}, \tag{4}$$

where $k_o$, $C_{cat}$, $C_{aro}$, $C_{bzh}$, $C_p$, $K_{aro}$, $K_{bzh}$ and $K_p$ represents reaction constant, the concentration of catalysts, aromatic, benzyl alcohol, mono benzylation product and adsorption coefficient of aromatic, benzyl alcohol mono benzylation product, respectively. Some side reactions like consecutive benzylation were not considered due to the high selectivity of Hβ-3 in mono benzylation products. It should be noted that in Langmuir–Hinshelwood mechanism, $k_0$ also includes adsorption coefficients,

$$k_0 = k'_0 K_{aro} K_{bzh} K_P \tag{5}$$

Based on the facts from experiment results that a superior selectivity towards mono benzylation products was observed under Hβ-3, to further investigate the adsorption procedure of different aromatic substrates and benzyl alcohol, quasi-initial conditions [54] were assumed to simplify Equation (4). The adsorption of product can be neglected in the very beginning of the reaction. The factor "1" in the denominator can be neglected because many works [54] have measured values of $K_{aro} C_{aro} + K_{bzh} C_{bzh}$ in similar conditions and found it in a much higher order of magnitudes than 1. So Equation (4) can be approximated as

$$r = \frac{k_0 C_{cat} C_{aro} C_{bzh}}{(K_{aro} C_{aro} + K_{bzh} C_{bzh})^2}. \tag{6}$$

We set $K_{bzh}/K_{aro}$ as x, and Equation (6) can be rewritten as

$$x^2 + \left(2 - \frac{k_0}{r}\right)\frac{c_{bzh}}{c_{aro}}x + \left(\frac{c_{bzh}}{c_{aro}}\right)^2 = 0, \tag{7}$$

$C_{bzh}/C_{aro}$ is constant in this paper, while $r = dC_{MBP}/dt$, the initial reaction rate can be obtained from the tangent slope at t = 0 min in Figure 14a. After solving this second-order Equation (7), the root at a meaningful range was selected. The resulting $K_{bzh}/K_{aro}$ ratios are listed in Table 4. On the one hand, it is very interesting that during initial benzylation of benzene, benzyl alcohol molecules adsorb 2.7 times faster than benzene molecules, and for the larger sized mesitylene molecules, the adsorption coefficient ratio of reactants is as large as 26 times. On the other hand, it is reported that the smaller capacity and longer equilibrium time of mesitylene adsorption make it not a good adsorbate candidate as smaller arenes like benzene [55,56]. So we predict the large arenes molecules occupy less active site than small ones, and more importantly, the benzylation agent molecules themselves have to compete with each other severely, because the opportunity to attack mesitylene molecule is far more less than attacking benzene in their respective benzylations.

**Table 4.** Kinetic results of adsorption and surface reaction in the benzylation of aromatics.

| Entry | Substrates | Kinetic Diameter [43,57] (Å) | $K_{BzH}/K_{are}$ | $k_a$ (min$^{-1}$) | $k_a{'}$ (mol·min$^{-1}$g$^{-1}$) |
|---|---|---|---|---|---|
| 1 | | 5.8 | 2.7 | 0.3993 | 2.9/K$^2_{BzOH}$ |
| 2 | | 6.7 | 7.2 | 0.1191 | 6.2/K$^2_{BzOH}$ |
| 3 | | 6.8 | 9.4 | 0.0566 | 5.0/K$^2_{BzOH}$ |
| 4 | | 8.6 | 26.0 | 0.0477 | 32/K$^2_{BzOH}$ |

$K_a{'}$ is the intrinsic surface reaction rate constant from Langmuir–Hinshelwood mechanism.

To the best of our knowledge, this is the first time that the steric effect in benzylation reactions was demonstrated from the view of competitive adsorption between guest molecules combining with macroscopic kinetic evidence. Vinu et al. [10] attributed the different apparent rate constant to the adsorption strength affected by increased electron density by more substituent group on benzene ring. However, they did not disentangle the measured rate constant from adsorption and adsorption rate is neglected. In contrast, Bachari et al. [58] explained the activity order using the Taft relation; they discovered the ionization potential of aromatic substrate obtained in literature had a linear relationship with lnK under Fe-HMS catalysts.

To demonstrate the etherification of benzyl alcohol, blank parallel experiments at 353 K and 373 K were employed: a mixture of 7 mL cyclohexane, 0.1 mL benzyl alcohol and 50 mg Hβ-3 was refluxed. At the beginning, BzOH converted to DBE rapidly, then the amount reached stable. The conversion of BzOH was fitted to second-order reaction ($R^2$ = 0.96) and $k_{ef}$ was measured as 0.0127 L/(mol·min) at 373 K (Figure 14b). It is worth to note the result in our study is quite different from Beltrame et al. [48], whose reaction order was fitted as -1 in the etherification of benzyl alcohol on Nafion−silica composite, and they met the form of equation of the Langmuir–Hinshelwood mechanism, so $K_{BzOH}$ was obtained through fitting the curve. In contrast, we did not obtain the $K_{BzOH}$, so we substituted $K_{BzOH}$ as a variable into Equation (5) and $k_a{'}$ was obtained in Table 4, assuming the same coefficient

$K_{BzOH}$ for each entry, the intrinsic surface reaction constant $k_a'$ shows the order: mesitylene > toluene > *p*-xylene > benzene, which is close to the classic view in electrophilic substitution, again proving the rationality of the macroscopic kinetic calculations here. Therefore, adsorption efficiency indeed played an important role in attributing to apparent rate constant and no wonder the so called steric effect dramatically influences green heterogeneous benzylations.

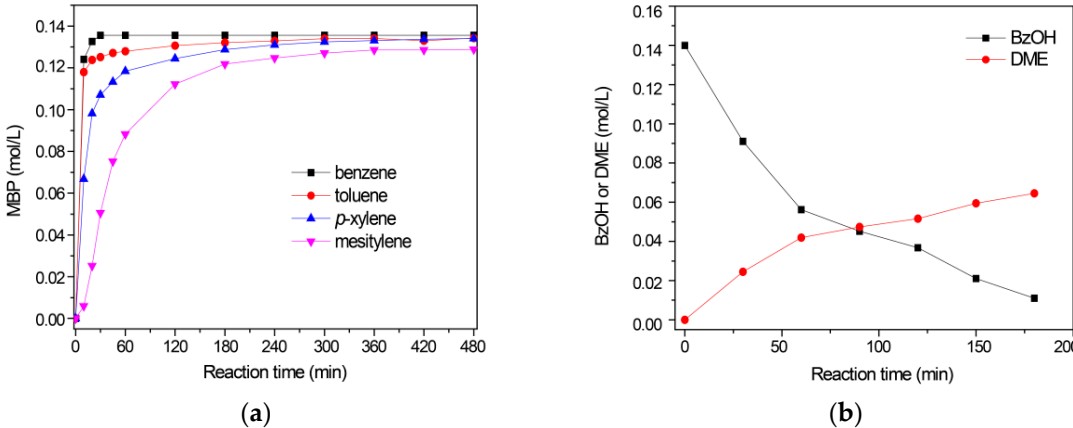

**Figure 14.** (**a**) The concentration of mono benzylation product (MBP) vs. reaction time. (**b**) The transformation of benzyl alcohol (BzOH) into dibenzyl ether (DBE) during etherification on Hβ-3.

## 3. Materials and Methods

### 3.1. Reagents

Reagents were obtained commercially and used without purification. Microporous zeolite (labeled as Hβ-1) with the specifications Si/Al ratio 25 was available from Nankai University Catalyst Plant, Tianjin, China. Polydiallyl-Dimethylammonium Chloride (PDADMA, molecular weight $1 \sim 2 \times 10^5$) was purchased from Aladdin Chemical Co., Ltd. (Shanghai, China)

### 3.2. Catalyst Preparation

Prior to use, all the catalysts were transformed into Hβ form through an ion-exchange procedure: 1g zeolites was stirred in 100 mL 1 mol/L ammonium nitrate solution at 358 K for 2 h. After filtrating and washing with purified water, the solid was dried at 383 K for 1h. This procedure was repeated for three times and the powders were calcined in air at 823 K for 5 h, and the ion-exchanged beta zeolite was labeled as Hβ-1.

Zeolites Hβ-2 and Hβ-3 were prepared based on Hβ-1 according the method reported in ref [28]. The above zeolite Hβ-1 was leached with a sodium hydroxide solution of 0.2 mol/L at 338 K for 0.5 h, and then the mixture was filtered and washed with purified water, dried, and ion exchanged three times into $NH_4$-form with the concentration of 1 mol/L ammonium nitrate, and then calcined at 823 K in air for 5 h. The acquired powder was transformed into H-form and labeled as Hβ-2.

A part of the Hβ-2 was then leached in a solution of 0.1 mol/L nitric acid at 358 K for 6 h (1 g zeolites stirred in 30 mL solution). After filtering, washing and drying in the same way mentioned above, the product was ion-exchanged into H-form and noted as Hβ-3. Sample Hβ-4 was synthesized using soft template PDADMA with a modified Si/Al ratio around 25 according to literature [30].

### 3.3. Characterizations

X-Ray Diffraction (XRD) tests were recorded by a Dandong Aolong Y-2000 diffractometer (Dandong, Liaoning., China) equipped with a Cu Kα monochromatized radiation source. The scanning speed is 0.05° per second. The relative crystallinity value was based on the strength of characteristic peak at 22.4°.

Nitrogen adsorption−desorption isotherms were measured on a static nitrogen adsorption system (JW-BK112 Beijing JEGB sci and tech. co., ltd., Beijing, China) to analyze the specific surface areas. The catalysts were degassed at 473 K under vacuum for 2 h to remove adsorbates before measurement.

X-ray fluorescence (XRF) measurement was conducted to analyze chemical composition using a PANalytical Axios-advance (Axios PW4400) spectrometer (Almelo, Netherlands). The Si/Al represents the mole ratio of $SiO_2$ to $Al_2O_3$.

Temperature-programmed desorption of ammonia ($NH_3$-TPD) was conducted in Micromeritics Autochem II 2920 (Atlanta, Georgia, USA) equipped with thermal conductivity detectors (TCD). About 100 mg of the sample was activated at 823 K under the flow of He for 1 h. After cooling to 373 K, the samples were saturated with $NH_3$ for 1 h followed by degassing to remove physically adsorbed $NH_3$. Then the desorption profiles were collected under a heating ramp of 10 K/min.

### 3.4. Benzylation of Arenes with Benzyl Alcohol

The catalyst was calcinated at 823 K for 3 h before use. The benzylation was carried out in a 25 mL two necks flask flowing nitrogen gas into the flask first to remove the water vapors before putting the catalyst into the flask, then 50 mg catalyst was added to the flask followed by 7 mL arene, and the flask was put into a temperature-controlled oil bath and heated homothermally under stirring at 900 r/min. 20 min later, 0.1 mL benzyl alcohol was added quickly to start the reaction. The recovered Hβ zeolites were activated in a tube furnace under a small quantity nitrogen flow at 723 K for 4 h, and then the zeolites cooled under flowing nitrogen atmosphere.

Reaction mixture (50 μL) of benzylation products were withdrawn periodically and dissolved in 1.00 mL ethanol for quantitative analysis by gas chromatography (Shimazu GC-2014 with a FID detector (Shimadzu, Nakagyo-ku, Kyoto, Japan) and capillary column) and a GC-MS Ultra equipped with a Rtx-5MS capillary column connected with a mass spectrometer (Agilent, Santa Clara, California, USA). Because of the excess of arene, the conversion was calculated based on benzyl alcohol. The selectivity was the amount of mono benzylation product divided by the total amount of products. The conversion, selectivity, and yield were calculated by:

$$Conversion\ \% = \frac{S_1 + S_2}{S_1 + S_2 + S_r} \times 100$$

$$Selectivity\ \% = \frac{S_1\ or\ S_2}{S_1 + S_2} \times 100$$

$$Yield\ \% = Conversion\ \% \times Selectivity\ \%,$$

where $S_1$ and $S_2$ were corresponding peak area of the products of benzylation and $S_r$ was the area of benzyl alcohol in GC analysis.

## 4. Conclusions

Hierarchical beta zeolites were applied in the catalytic benzylation of arenes with benzyl alcohol. The basic structures of prepared hierarchical Hβ zeolites were confirmed by XRD and $N_2$ sorption. It was found that the Hβ-3 catalyst prepared by post-treatment method had the highest selectivity and catalytic activity in the probe reaction test. The benzylation of aromatics with different molecular sizes was evaluated respectively. The activities of aromatic substrates follow the order: benzene > toluene > *p*-xylene > mesitylene. The influence of reaction temperatures was also studied. As the temperature increased, both the conversion of raw material and selectivity of main product increased rapidly, and the overall apparent activated energy in mesitylene benzylation was measured as 169 kJ/mol.

Based on the above systematic study, the reaction mechanism was further discussed. Different from previously reported results, the undesired etherification of benzyl alcohol on Hβ-3 could be fitted into second-order reaction. For the benzylation of arenes, the whole reaction process was divided into five steps to illustrate the steric effect in detail. Estimated by Mears' criteria and Weisz–Prater' criteria,

diffusion is unlikely to be the controlled step though the arenes showed certain differences in diffusion. With the assumption of quasi-initial conditions under the Langmuir–Hinshelwood mechanism, the steric effect was linked with different adsorption coefficient ratio of reactants. The intrinsic reaction order is in accordance with classical electrophilic substitution rules. This work may give new insights in developing green catalytic processes for heterogeneous alkylation.

**Author Contributions:** Data curation, X.L, M.L. and X.W.; writing—original draft, X.L.; writing—review and editing, J.L. and M.L.; project supervision, J.Y. The manuscript has been read and revised by all authors before submission.

**Funding:** This research was funded by the National Natural Science Foundation of China (51663009), the science and technology planning project of Haikou (2016030), the Exploration planning on Teaching Reform in Higher Education Institutions of Hainan Province (Hnjg 2016ZD-5) and Hainan University student innovation and entrepreneurship fund project (Hys 2018-78).

**Acknowledgments:** Yongming Zhang, Qinhe Pan, Jun Hu, Shuai Yu, Jiangwen Zou, and the Analytical and Testing Center of Hainan University are acknowledged for their help in fruitful discussions and material characterizations.

**Conflicts of Interest:** The authors declare no conflict of interest.

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
