# Peer review of "The Steric Effect in Green Benzylation of Arenes with Benzyl Alcohol Catalyzed by Hierarchical H-beta Zeolite"

_catalysts, doi:10.3390/catal9100869_

Round 1

Reviewer 1 Report

The subject of the paper is quite of a mature nature. Therefore, the authors should have surprised the readers with the originality in preparation or catalytic application.

The following comments should be taken into account in the revision of the manuscript:

1. The Si/Al ratios for H-Beta-2 and H-Beta-3 samples should be indicated. If they are different from those of H-Beta-1 and H-Beta-4, then the comparison does not make sense.

2. Figures 3-9, Tables 2,3: the conversion, selectivity and yield should be related to the certain compound – conversion of … (especially taking into account that two liquid substrates are used), selectivity to … and yield of …. These compounds should be indicated in the captions/legend.

3. The yields of disubstituted products should be indicated.

4. The authors overused the word “later” and usually not in the proper meaning.

5. The substantial improvement of the English language usage is required.

Reviewer 2 Report

In this manuscript, the authors prepared mesoporous aluminosilicate *BEA zeolites with different textural properties, which were applied to benzylation reactions. Several technique were used to confirm the formation of mesoporous zeolites and results of catalytic tests were consistent each other. The understanding for the reaction mechanism and the assumptions made in this study were, however, too jumpy and lacked several important information. Following points should be clearly answered in the manuscript.

Major comment 1:

Si/Al ratio of Hβ-2,3,4 should be experimentally measured. As the authors mentioned, the reaction is catalyzed by acid site. The relative number of Al site per weight of catalyst should be changed by the post-treatment, because the post-treatment is a desilication process. 

Furthermore, I wonder how the catalytic performance changes if authors compare the result (Fig.3) based on the number of Al sites. 

Major comment 2:

In the section 2.5, the authors mixed the diffusion into a reaction mechanism. In general, "reaction scheme" only describes elementary steps of the reaction, and that does not contain diffusion of the molecule towards the catalytic site. Actually, "Diffusion" step described in Fig.12 is "adsorption".  And as far as the discussion in section 2.5.2 might be acceptable, it should be discussed separately from "reaction mechanism" or "reaction path". Furthermore, in Fig.12, the description of H-β is sometimes weird. How is "H" on zeolite regenerated in the cycle? Together with minor comment 3, Fig.12 and related discussions should be totally renewed.

Major comment 3:

As the author mentioned in the literature, in my understanding based on the manuscript (especially Fig.5-9 and line 209-211), the main path to form the final product (DPM, BMB, BDB, and BTB) is benzylation of ether intermediate (DBE) and aromatic reactant. (Please show a plot about selectivity to DBE and DPM (y-axis) along with reaction time (x-axis).)

If this is true, DBE seems to be larger than aromatic reactants. Then, why did the steric hindrance of (smaller) aromatic reactant affected the reaction? 

If this is not, please show some evidence or provide discussion that the major path is the direct benzylation reaction.

Major comment 4:

This is also related to the major comment 3. In the section 2.5.3, authors assumed the L-H model. Although it might be fine, the equation 4 only assumed one of the reaction path (direct benzylation). Why was the other path (DBE + aromatics) avoided? The reason was not clear. Furthermore, in line 273, authors assumed that the surface sites were saturated by reactants. Do you have any reason for this assumption? 

Minor comment 1:

In table 2, the letters "a", "b", and "c" should be clarified. And, how is the "ka" value calculated? 

Minor comment 2:

In the section 2.4, how much did you collected the catalyst at each cycle? The loss of catalyst during filtering procedures should be clarified. 

Minor comment 3:

In figure 12, chemical structure of DBE is incorrect. 

Minor comment 4:

In lines 233 and 241, what is the difference between ρb (bulk density of catalyst) and ρcat (density of catalyst)? Why these values are different? 

Reviewer 3 Report

In this work, Liu et al. report an experimental investigation on the application of hierarchical BEA zeolite in benzylation of arenes with benzyl alcohol. Higher activity was exhibited by hierarchical sample prepared by sequential desilication and leaching, and this aspect was deeply discussed. The manuscript is well written and the results interesting. Therefore, the manuscript is suitable for publication after minor revision. Herewith, the authors may find my comments:

The Si/Al ratio of all the samples shall be measured and reported in Table 1; The t-plot method may have some problems in the estimation of micropore volume for micro/meso materials. The following studies may be useful and should be cited: Lanzafame et al., Appl. Catal A., 580 (2019) 186-196; Pérez-Ramirez et al., Chem. Soc. Rev. 37 (2008) 2530-2542; Galarneau et al., Langmuir 30 (2014) 13266-13274. PSD should be reported; The N2 isotherms should be better discussed at the light of the above suggested references. For instance, nitrogen is not a suitable probe molecule for zeolites and it should be highlighted.

The paper is focused on the utilization of hierarchical zeolites, therefore these aspects are of paramount importance.

Round 2

Reviewer 2 Report

The revised manuscript carefully addressed the concerns I mentioned in the first round. It would be acceptable in the current form. 

Author Response

We would like to thank the Reviewer for his/her decision.